# BMP9-ID1 Pathway Attenuates N^6^-Methyladenosine Levels of CyclinD1 to Promote Cell Proliferation in Hepatocellular Carcinoma

**DOI:** 10.3390/ijms25020981

**Published:** 2024-01-12

**Authors:** Han Chen, Mingming Zhang, Jianhao Li, Miao Liu, Dan Cao, Ying-Yi Li, Taro Yamashita, Kouki Nio, Hong Tang

**Affiliations:** 1Center of Infectious Diseases, West China Hospital of Sichuan University, Chengdu 610041, China; chenhan19890801@wchscu.cn (H.C.); zhangmm@stu.scu.edu.cn (M.Z.); 2022324025209@stu.scu.edu.cn (J.L.); liumiao@wchscu.cn (M.L.); 202025206@alu.scu.edu.cn (D.C.); 2Division of Infectious Diseases, State Key Laboratory of Biotherapy and Center of Infectious Diseases, West China Hospital of Sichuan University, Chengdu 610041, China; 3Department of Gastroenterology, Kanazawa University Hospital, Kanazawa 9208641, Japan; liyingyi@staff.kanazawa-u.ac.jp (Y.-Y.L.); taroy62m@staff.kanazawa-u.ac.jp (T.Y.)

**Keywords:** BMP9-ID1 pathway, m^6^A methylation, FTO, CyclinD1, hepatocellular carcinoma, BMP receptor inhibitor

## Abstract

Hepatocellular carcinoma (HCC) is a highly lethal malignant neoplasm, and the involvement of bone morphogenetic protein 9 (BMP9) has been implicated in the pathogenesis of liver diseases and HCC. Our goal was to investigate the role of BMP9 signaling in regulating N6-methyladenosine (m^6^A) methylation and cell cycle progression, and evaluate the therapeutic potential of BMP receptor inhibitors for HCC treatment. We observed that elevated levels of BMP9 expression in tumor tissues or serum samples from HCC patients were associated with a poorer prognosis. Through in vitro experiments utilizing the m^6^A dot blotting assay, we ascertained that BMP9 reduced the global RNA m^6^A methylation level in Huh7 and Hep3B cells, thereby facilitating their cell cycle progression. This effect was mediated by an increase in the expression of the inhibitor of DNA-binding protein 1 (ID1). Additionally, using methylated RNA immunoprecipitation qPCR(MeRIP-qPCR), we showed that the BMP9-ID1 pathway promoted CyclinD1 expression by decreasing the m^6^A methylation level in the 5′ UTR of mRNA. This occurred through the upregulation of the fat mass and obesity-associated protein (FTO) in Huh7 and Hep3B cells. In our in vivo mouse xenograft models, we demonstrated that blocking the BMP receptor with LDN-212854 effectively suppressed HCC growth and induced global RNA m^6^A methylation. Overall, our findings indicate that the BMP9-ID1 pathway promotes HCC cell proliferation by down-regulating the m^6^A methylation level in the 5′ UTR of CyclinD1 mRNA. Targeting the BMP9-ID1 pathway holds promise as a potential therapeutic strategy for treating HCC.

## 1. Introduction

The global impact of hepatocellular carcinoma (HCC), ranking as the second most prevalent cause of cancer, is linked with substantial morbidity and mortality [1]. The intractability of HCC partly stems from the limited availability of effective pharmacological treatments [2,3]. Bone Morphogenetic Proteins (BMPs), a subgroup of signaling molecules within the Transforming Growth Factor-beta (TGF-β) superfamily, hold a pivotal role in liver development [4]. Among the BMP family members, BMP9 has recently gained attention due to its involvement in stem cell differentiation, angiogenesis, metabolism, fibrosis, and tumor growth within the liver [5]. In our published studies, we provided evidence of a significant correlation between increased BMP9 expression and adverse prognosis in HCC. Additionally, we explicated the role of BMP9 in advocating cancer stem cell properties, as indicated by Epithelial Cellular Adhesion Molecule (EpCAM) positivity. We also discovered that BMP9 facilitates angiogenesis through Hypoxia-Inducible Factor 1 (HIF-1α) and Vascular Endothelial Growth Factor A (VEGF-A) signaling by restraining the activity of DNA-binding protein 1 (ID1) in HCC. However, further investigation is needed to clarify additional mechanisms underlying the regulation of HCC development by BMP9-ID1 pathway.

Eukaryotic RNA undergoes more than 150 biological modifications. Methylation is widely distributed across various types of RNA, including messenger RNA (mRNA), transfer RNA (tRNA), ribosomal RNA (rRNA), small non-coding RNA, and long non-coding RNA (lncRNA) [6]. Reported RNA methylation modifications include N6-methyladenosine (m^6^A), 5-methylcytosine (m^5^C), N1-methyladenosine (m^1^A), N7-methylguanosine (m^7^G), N4-acetylcytosine (ac^4^C), pseudouridine (Ψ), uridylation, and adenosine-to-inosine (A-to-I) RNA editing [7,8,9,10,11,12]. In eukaryotes, methylation in mRNA is a prevalent and critical post-transcriptional modification. It plays key roles in various aspects of mRNA metabolism and function, such as regulation of mRNA stability and degradation, influence on translation efficiency, alternative splicing, nuclear export, and mRNA folding [13,14]. Similarly, methylation in tRNAs is a common and vital post-transcriptional modification, with modifications occurring at various positions within the molecule, including the base, ribose, or phosphate backbone. Each of these modifications potentially carries out different functions [15,16]. Methylation also plays essential roles in rRNA maturation and function. It is involved in ribosome biogenesis, stabilization of ribosome structure, translation efficiency and fidelity, interactions with tRNAs and mRNAs, as well as response to cellular stress [17]. Recent studies highlight the presence of m^6^A methylation in primary microRNAs (pri-miRNAs), affecting their processing into mature miRNAs and thereby influencing miRNA-mediated gene silencing [18]. In lncRNAs, m^6^A methylation can influence their stability by affecting their degradation rate. It can also alter the secondary structure of lncRNAs, potentially influencing their interactions with other biomolecules and impacting their functions [19,20].

With m^6^A methylation being the most prevalent among them, RNA m^6^A methylation modification is catalyzed by m^6^A “writers” (methyltransferases) and can be reversed by m^6^A “erasers” (demethylases), constituting a dynamic and reversible process [6,21]. The mRNA bearing m^6^A methylation modification is recognized by m^6^A “readers” (methylated RNA-binding proteins) and actively participates in downstream processes of RNA translation and degradation [22,23]. The known writers include METTL3, METTL14, METTL16, WTAP, KIAA1429, RBM15, and ZC3H13; the known erasers consist of FTO and ALKBH5; while the known readers comprise YTHDF1-3, IGF2BP1-3, YTHDC1, HNRNPG, and eIF3A. The translation process involves YTHDF1, YTHDF3, YTHDC2, IGF2BP1-3, METTL3, and eIF3. Additionally, the stability of mRNA is regulated by YTHDF2, YTHDF3, YTHDC2, and IGF2BP1-3, with YTHDF2, YTHDF3, and YTHDC2 associated with mRNA decay, and IGF2BP1-3 playing a role in mRNA stabilization. The YTHDC1 protein is involved in the regulation of splicing events and facilitates nuclear export processes of mRNA [24,25]. m^6^A methylation plays a pivotal role in the pathogenesis and progression of liver diseases and HCC, involving multifactorial mechanisms and intricate signaling networks, thus warranting further investigation [26,27,28]. In this study, we elucidate the potential role of BMP9-ID1 pathway in modulating cell proliferation and m^6^A methylation in HCC cells. Concurrently, we underscore the antitumorigenic impact brought about by BMP receptor inhibitors via the suppression of the BMP9-ID1 pathway.

## 2. Results

### 2.1. BMP9 Expression Is Related to the Expression of CyclinD1 in HCC Tissues

To investigate the correlation between BMP9 expression in HCC tissue specimens and patient prognosis, we conducted an analysis of BMP9 expression in HCC specimens of 51 HCC patients using immunohistochemical (IHC) staining. Patients with higher BMP9 expression in HCC tissue than in the adjacent liver tissue were categorized as BMP9-high, while those with the same or lower BMP9 expression in HCC tissue were categorized as BMP9-low (Figure 1A). The results of IHC staining revealed elevated 26 patients were in BMP9-high group and 25 patients were in BMP9-low group. To further elucidate the prognostic significance of BMP9 expression in HCC patients, we categorized these 51 patients into two groups based on the differential expression of BMP9 between tumor tissue and adjacent liver tissue and subsequently assessed their overall survival. Interestingly, patients with high BMP9 expression (n = 26) in HCC demonstrated significantly poorer overall survival compared to those with low BMP9 expression (n = 25) (*p* = 0.0142, Figure 1B and Appendix A). The data presented herein suggest that elevated BMP9 expression in tumor tissue is significantly associated with unfavorable overall survival and may serve as a valuable prognostic biomarker for HCC. Furthermore, the investigation of clinical characteristics in these patients revealed a significant correlation between elevated BMP9 expression in HCC tissues and increased serum AFP levels (Table 1 and Appendix A).

Cyclin D1, encoded by the *CCND1* gene, is one of the most important regulators in regulating the cell cycle and serves as a significant prognostic and predictive factor across various cancers [29,30,31]. We examined the CyclinD1 expression in HCC specimens using IHC. Patients with higher CyclinD1 expression in HCC tissue than in the adjacent liver tissue were categorized as CyclinD1-high (CyclinD1+), while those with the same or lower CyclinD1 expression in HCC tissue were categorized as CyclinD1-low (CyclinD1-).

By conducting further analysis on the correlation between BMP9 and Cyclin D1 in HCC tissue specimens, we discovered a significant association between BMP9 expression and Cyclin D1 expression (*p* = 0.0087, Figure 1C,D). To further validate the association between BMP9 and Cyclin D1, we utilized data from the GEPIA2 dataset (http://gepia2.cancer-pku.cn/#index, accessed on 15 September 2022), which originated from the TCGA-LIHC datasets. The correlation analysis of BMP9 and Cyclin D1 aligned with our tissue data (Figure 1E). The above results suggest that BMP9 may be involved in regulating the cell cycle progression of HCC.

### 2.2. BMP9 Enhances CyclinD1 Expression in HCC Cells to Facilitate Cell Cycle Progression via Suppressing m^6^A Methylation within the 5′-UTR of CyclinD1 mRNA

The role of BMP9 in HCC cell proliferation was subsequently investigated. Initially, we examined the impact of recombinant human BMP9 on the cell proliferation of Huh7 and Hep3B cell lines. The CCK-8 and colony formation assays unequivocally demonstrated that BMP9 significantly enhanced cellular proliferation in these HCC cell lines compared to the control group (Figure 2A,B). To further evaluate the effect of BMP9, we assessed the cell cycle distribution by flow cytometry. Notably, treatment with BMP9 resulted in a significant reduction in the G1 phase and an increase in the proportion of cells in the S phase for both Huh7 and Hep3B cells. These findings suggest that BMP9 promotes the transition from G1 to S phase during cell cycle progression in HCC cells (Figure 2C). m^6^A dot blotting is a commonly used technique for the detection global levels of m^6^A modification in RNA samples. We measured the global m^6^A levels in both the negative control and BMP9-treated Hep3B and Huh7 cell lines using m^6^A dot blotting. Remarkably, treatment with BMP9 led to a substantial reduction in global m^6^A methylation levels in both Huh7 and Hep3B cell lines (Figure 2D). In order to elucidate the impact of BMP9 on proteins associated with m^6^A methylation modification, we comprehensively investigated all known writers, erasers, and readers involved in m^6^A methylation modification via RT-qPCR. The results revealed variations in the expressions of writers, which posed challenges for analysis (Appendix A). Among the erasers, only the expression of FTO increased, while no significant alteration was observed in the expression of ALKBH5 (Appendix A). Regarding readers, the expression of YTHDF2, a regulator of mRNA decay, was found to be down-regulated, whereas the expression of IGF2BP1-3, regulators of mRNA stability, exhibited up-regulation (Appendix A). Consequently, our subsequent research will primarily focus on investigating the roles of FTO and YTHDF2. Furthermore, RT-qPCR and Western blotting analyses showed that BMP9 significantly up-regulated the expression of ID1, CyclinD1, and FTO, while downregulating YTHDF2 at both the mRNA and protein level (Figure 2E,F). It has been reported that Cyclin D1 contains m^6^A sites in both the 5′ UTR and 3′ UTR, with a significantly higher abundance observed in the 5′ UTR compared to the 3′ UTR [32]. To specifically detect the 5′ UTR region of CyclinD1, we utilized RT-qPCR primers designed to target this specific region of CyclinD1. The MeRIP-qPCR analysis revealed a significant reduction in m^6^A methylation within the 5′-UTR of CyclinD1 in Huh7 and Hep3B cell lines upon BMP9 treatment. These findings suggest that BMP9 enhances HCC cell proliferation by promoting G1/S phase transition through the up-regulation of CyclinD1, achieved via the reduction of m^6^A methylation within the 5′ UTR region of CyclinD1.

### 2.3. The BMP9-ID1 Pathway Facilitates HCC Cell Cycle Progression by Suppressing m^6^A Methylation within the 5′ UTR of CyclinD1 mRNA

We undertook a study to decipher the mechanism through which BMP9 triggers the expression of CyclinD1. Since ID1 is a direct target of BMP9, we hypothesized that ID1 plays a regulatory role in the BMP9-induced expression of CyclinD1 and FTO. The same as BMP9 and CyclinD1, we defined patients with higher ID1 expression in HCC tissue than in the adjacent liver tissue were categorized as ID1-high (ID1+), while those with the same or lower ID1 expression in HCC tissue were categorized as ID1-low (ID1-).

Our expectations were confirmed through IHC staining analysis of HCC tissues, which revealed a significant correlation between CyclinD1 and ID1 expression levels (*p* = 0.0193, Figure 3A,B). The correlation analysis data obtained from the GEPIA2 dataset (15 September 2022) also supported our anticipated results (Figure 3C).

In Huh7 and Hep3B cells overexpressing ID1, there was a significant enhancement in cell proliferation observed in Huh7 and Hep3B cell lines (Figure 4A,B). Flow cytometry analysis showed that ID1 overexpression promoted G1/S phase transition in Huh7 and Hep3B cell lines (Figure 4C). Additionally, overexpression of ID1 was found to suppress the global RNA m^6^A methylation in Huh7 and Hep3B cell lines (Figure 4D). RT-qPCR and Western blotting analyses demonstrated that ID1 upregulated the expression of CyclinD1, and FTO, while simultaneously downregulating YTHDF2 at both mRNA and protein levels (Figure 4E,F). MeRIP-qPCR analysis revealed a significant reduction in m^6^A methylation within the 5′ UTR of CyclinD1 mRNA in Huh7 and Hep3B cell lines upon ID1 overexpression (Figure 4G).

On the other hand, ID1 knockdown inhibited cell proliferation in Huh7 and Hep3B cell lines (Figure 5A,B). Flow cytometry analysis revealed that ID1 knockdown hindered G1/S phase transition in Huh7 and Hep3B cell lines (Figure 5C). Furthermore, knockdown of ID1 significantly increased global RNA m^6^A methylation in Huh7 and Hep3B cell lines (Figure 5D). RT-qPCR and Western blotting analyses demonstrated that knockdown of ID1 led to a significant reduction in the expression CyclinD1 and FTO, while simultaneously inducing the expression of YTHDF2 at both mRNA and protein levels (Figure 5E,F). MeRIP-qPCR analysis revealed a significant induction in m^6^A methylation within the 5′ UTR of CyclinD1 mRNA in Huh7 and Hep3B cell lines upon ID1 overexpression (Figure 5G).

Moreover, we validated that knockdown of ID1 significantly attenuated BMP9-induced cellular proliferation in Huh7 and Hep3B cells (Figure 6A,B). The knockdown of ID1 also mitigated the BMP9-induced G/S1 phase translation of Huh7 and Hep3B cells (Figure 6C). The m^6^A dot blotting assay revealed that knockdown of ID1 resulted in a decrease in the inhibitory effect on global RNA m^6^A methylation induced by BMP9 (Figure 6D). Additionally, knockdown of ID1 attenuated the BMP9-induced upregulation of CyclinD1 and FTO, as well as the downregulation of YTHDF2 expression (Figure 6E,F). Furthermore, MeRIP-qPCR analysis indicated that knockdown of ID1 effectively attenuated the BMP9-induced decrease in m^6^A methylation within the 5′ UTR of CyclinD1 mRNA in Huh7 and Hep3B cell lines (Figure 6G). Taken together, these findings suggest that BMP9-ID1 facilitates HCC cell proliferation and G1/S phase transition by suppressing m^6^A methylation within the 5′ UTR of CyclinD1 mRNA.

### 2.4. BMP9-ID1 Pathway Promotes CyclinD1 Expression via FTO

Since FTO is the only reader that can be positively regulated by BMP9 (Appendix A), we hypothesize that BMP9-ID1 pathway regulates CyclinD1 expression through FTO. As anticipated, when FTO was knocked down, the upregulation of CyclinD1 induced by BMP9 was inhibited (Figure 7A,B). Furthermore, the knockdown of FTO completely abolished the upregulation of CyclinD1 triggered by ID1 (Figure 7C). These findings strongly suggest that FTO plays a vital role as a downstream target of BMP9-ID1 pathway in the progression of HCC.

### 2.5. BMP Receptor Inhibitors Attenuate Upregulation of Progression of Cell Cycle and Downregulate m^6^A Methylation within the 5′ UTR of CyclinD1 mRNA Induced by BMP9 in HCC Cells

The aforementioned data strongly suggests that targeting the BMP9-ID1 pathway could serve as a crucial therapeutic strategy for suppressing the development of HCC. Consequently, we continued to explore the impact of BMP receptor inhibitors K02288 and LDN-212854 on HCC. In comparison to the control group, inhibition of BMP receptors significantly attenuated cell proliferation in Huh7 and Hep3B cells (Figure 8A,B). Furthermore, BMP receptor inhibition resulted in the suppression of G1/S phase transition in Huh7 and Hep3B (Figure 8C). The m^6^A dot blotting assay revealed a significant increase in global RNA m^6^A methylation in Huh7 and Hep3B cell lines upon treatment with BMP receptor inhibitors (Figure 8D). Additionally, suppression of BMP receptors resulted in a reduction in the expression levels of ID1, CyclinD1, and FTO, while simultaneously increasing the expression of YTHDF2 (Figure 8E,F). Moreover, the inhibition of BMP receptors induced m^6^A methylation within the 5′ UTR of CyclinD1 mRNA (Figure 8G).

To further assess the influence of BMP receptor inhibitors in the context of BMP9, we hypothesized that K02288 and LDN-212854 would substantially impede the proliferation induced by BMP9 and G1/S phase transition in Huh7 and Hep3B cells (Figure 9A–C). The inhibition of global RNA m^6^A methylation induced by BMP9 was also mitigated by BMP receptor inhibitors in Huh7 and Hep3B cells (Figure 9D). Furthermore, BMP receptor inhibitors inhibited the upregulation of ID1, CyclinD1, and FTO induced by BMP9, while attenuating the downregulation of YTHDF2 induced by BMP9 (Figure 9E,F). Moreover, the inhibition of BMP receptors effectively eliminated the suppressive effect of BMP9 on m^6^A methylation within the 5′ UTR of CyclinD1 mRNA. In summary, these findings suggest that BMP receptor inhibitors harbor potential as prospective therapeutic agents for impeding the progression of HCC.

### 2.6. BMP9 Receptor Inhibitor Suppresses HCC Tumor Growth and CyclinD1 Expression While Inducing Global RNA m^6^A Methylation In Vivo

Based on the in vitro findings, LDN-212854 demonstrated the highest potency in suppressing the progression of HCC. Consequently, we selected LDN-212854 as the inhibitor to evaluate its antitumor effect on HCC xenografts. Compared to PBS, LDN-212854 considerably inhibited tumor growth derived from Huh7 and Hep3B cells in an in vivo mouse xenograft model (Figure 10A,B). Additionally, LDN-212854 was found to induce global RNA m^6^A methylation in xenograft tumor tissues (Figure 10C). Consistent with the in vitro results, Western blot, RT-qPCR, and IHC staining revealed that LDN-212854 effectively inhibited the expression of ID1, CyclinD1, and FTO in vivo, while notably up-regulating the expression of YTHDF2 (Figure 10D–F). These results demonstrate the effective suppression of HCC tumor growth and CyclinD1 expression by the BMP receptor inhibitor LDN-212854. Simultaneously, it promotes global RNA m^6^A methylation levels in HCC through the inhibition of BMP9-ID1 pathway. These findings suggest that targeting the BMP9-ID1 pathway holds tremendous promise as a therapeutic strategy for patients with HCC.

## 3. Discussion

The BMP9-ID1 pathway has been implicated in the development and progression of HCC. In our recent studies, we have identified that BMP9 plays a pivotal role in enhancing the malignancy of HCC. Our findings suggest that the BMP9-ID1 pathway fosters cancer stem cell characteristics in EpCAM-positive HCC cells by triggering the Wnt/β-catenin signaling pathway. Furthermore, we unveiled that the BMP9-ID1 pathway propels angiogenesis in HCC via the activation of HIF-1α/VEGFA. The mechanism by which BMP9 regulates the occurrence and development of liver cancer is complex, involving multiple signaling pathways. Therefore, further investigation into this area is necessary and valuable [33,34].

Post-transcriptional regulatory mechanisms in tumor biology, particularly m^6^A modification, have been extensively studied. m^6^A modification plays a pivotal role in various cellular metabolic processes, including RNA translation, decay, and structural transformations. It has emerged as an intriguing research focus in recent years. Bioinformatic analyses have unveiled significant correlations between YTHDF1/2, YTHDC1, RBM15, and METTL3 with the clinical stages of HCC. A decrease in METTL14 expression, accompanied by an increase in other m^6^A regulators, has been linked to a poorer prognosis. Additionally, the expression of the YTHDF family displayed a significant correlation with immune infiltration within the liver cancer micro-environment [35]. For instance, m^6^A methyltransferase METTL3 promotes NAFLD-HCC [36], while METTL14 curbs the metastasis of HCC by regulating the EGFR/PI3K/AKT signaling pathway in a manner that depends on m^6^A methylation [37]. FTO has been shown to play an important role in HCC progression through the promotion of hepatic inflammation and cancer stemness [38,39,40]. However, there have been conflicting reports on the HCC protective effect of FTO [41,42,43]. Among m^6^A readers, the overexpression of YTHDF1 correlates with a poorer prognosis in patients with HCC [44]. YTHDF1 promotes the progression of HCC by stimulating the PI3K/AKT/mTOR signaling pathway and inducing epithelial-mesenchymal transition, thereby promoting oncogenicity [45,46]. Conversely, YTHDF2 inhibits cell proliferation and growth in HCC by destabilizing EGFR mRNA [47], while also reducing inflammation and vascular abnormalities in HCC [48]. However, several studies have reported the oncogenic properties of YTHDF2 [49,50]. All these studies demonstrate the involvement of m^6^A regulators in HCC, highlighting the importance of investigating upstream regulators of m^6^A methylation to understand the comprehensive mechanisms underlying its role in HCC.

In our study, we elucidated a novel mechanism by which the BMP9-ID1 pathway regulates HCC cell cycle progression through m^6^A methylation of CyclinD1. Initially, we demonstrated the association between BMP9 and CyclinD1 by analyzing HCC tissue specimens and data from the TCGA-LIHC datasets, suggesting the involvement of BMP9 in HCC development. Consistent with this concept, we validated the induction of ID1 and CyclinD1 expression in HCC cells when administering human recombinant BMP9 protein, leading to the facilitation of G1/S phase cell cycle progression. Considering the crucial role of m^6^A methylation in the progression of HCC, we conducted further investigations to uncover the mechanistic link between BMP9 and CyclinD1 expression through m^6^A methylation-related pathways. The m^6^A methylation level is collectively determined by the methyltransferase, demethylase, and methyl-binding protein. Considering the variability in expressions among writers, with some showing an increase while others a decrease, it remains challenging to draw definitive conclusions. Therefore, we focused our investigation on erasers and readers. Notably, BMP9 selectively modulated FTO expression without affecting ALKBH5. The decay of m^6^A methylated mRNA is primarily regulated by YTHDF2 and YTHDF3.

In this study, we elucidated a novel mechanism by which the BMP9-ID1 pathway regulates HCC cell cycle progression through m^6^A methylation of CyclinD1 (Figure 11). We initially demonstrated the association between BMP9 and CyclinD1 by analyzing HCC tissue specimens alongside data obtained from the GEPIA2 dataset. These findings suggest the involvement of BMP9 in HCC development. In line with this concept, we validated the induction of ID1 and CyclinD1 expression in HCC cells by administering recombinant BMP9 protein, leading to the facilitation of G1/S phase cell cycle progression. Given the crucial role of m^6^A methylation in HCC development, we conducted further investigations to elucidate the mechanistic link between BMP9 and CyclinD1 expression through m^6^A methylation-related pathways. The m^6^A methylation level is collectively determined by the methyltransferase (writers), demethylase (erasers), and methyl-binding protein (readers). Given the variability in expressions among writers, with some showing an increase while others a decrease, making it challenging to draw definitive conclusions, therefore we focused our investigation on erasers and readers. It is noteworthy that BMP9 exclusively modulated FTO without affecting ALKBH5. It is already known that the decay of m^6^A methylated mRNA is predominantly regulated by YTHDF2 and YTHDF3 [51,52]. However, our findings indicate that BMP9 selectively downregulates YTHDF2 expression without affecting YTHDF3 expression. Furthermore, our findings demonstrate that ID1 plays a role in upregulating the expression of CyclinD1 and FTO, promoting cell cycle progression from G1 to S phase. Simultaneously, ID1 downregulates the expression of YTHDF2 and m^6^A methylation levels in HCC cells. Knockdown of ID1 also attenuated the effects of BMP9. Moreover, both in vitro and in vivo experiments revealed that BMP receptor inhibitors effectively suppressed HCC cell proliferation and cell cycle progression. Additionally, they downregulated the expression levels of ID1, FTO, and CyclinD1. Notably, BMP receptor inhibitors promoted m^6^A methylation levels while enhancing YTHDF2 expression.

## 4. Materials and Methods

### 4.1. Clinical Samples

A collection of 51 resected HCC tissue samples (Appendix A) was acquired following informed consent from patients who underwent liver resection at the Western China Hospital of Sichuan University between 2018 and 2019. Informed consent for inclusion was procured from all subjects before their participation in the study, which would utilize the tissues for future research. The study adhered to the principles outlined in the Declaration of Helsinki, and the protocols were sanctioned by the Ethics Committee of the Western China Hospital of Sichuan University (2016-91).

### 4.2. Cell Lines and Reagents

The HCC cell lines Huh7 and Hep3B were procured from Procell Life Science & Technology (Wuhan, China). The HCC cells were cultivated in Dulbecco’s Modified Eagle Medium (DMEM; Gibco, Grand Island, NY, USA) that was supplemented with Fetal Bovine Serum (FBS; Gibco), and incubated at 37 °C in an atmosphere of 5% CO_2_.

### 4.3. RNA Interference and Plasmid Transfection

The ID1-specific siRNA and the negative control siRNA were obtained from GenePharma (Shanghai, China). Prior to siRNA transfection, cells were cultured in FBS-free medium for 24 h. The siRNA constructs were introduced using Lipofectamine RNAiMAX (Invitrogen, Waltham, MA, USA) following the manufacturer’s guidelines. Approximately 4–6 h post-transfection, the cells were rinsed with PBS to ensure total removal of any residual siRNA constructs in the medium, after which they were cultured in DMEM supplemented with 10% FBS. The PCMV6-AC-GFP-ID1 (RG202061) was procured from Origene Technologies, Inc. (Rockville, MD, USA), while the pcDNA3.1 (V790-20) plasmid, utilized as an empty vector control, was bought from Invitrogen. DNA constructs were transfected employing Lipofectamine 3000 (Invitrogen), adhering to the manufacturer’s protocol. The co-transfection of siRNA and plasmid was executed with Lipofectamine 2000 (Invitrogen) in line with the manufacturer’s instructions. About 4–6 h subsequent to transfection, the cells were washed with PBS to entirely eliminate the DNA constructs from the medium, followed by culturing in DMEM enriched with 10% FBS.

### 4.4. Cell Proliferation Assay and Colony Formation

Cell proliferation was evaluated using a Cell Counting Kit-8 (CCK-8, BIMAKE, Houston, TX, USA). Each well of a 96-well plate was seeded with 2 × 10^3^ cells in triplicate and incubated for one hour at 37 °C. Absorbance was then measured at 450 nm daily for three consecutive days. For the colony formation assay, 2 × 10^3^ treated cells were plated into 6-well plates in triplicate. After an incubation period of 14 days, these plates underwent two washes with phosphate-buffered saline (PBS), followed by a fixation with 4% formaldehyde for 10 min. Subsequently, cells were stained with a 0.1% crystal violet solution (Cat#G1064, Solarbio, Beijing, China) for 10 min, enabling further analyses.

### 4.5. Cell Cycle Analysis

Cells were harvested and fixed using chilled 70% ethanol and left at 4 °C overnight. Once the ethanol was removed, cells underwent two washes with PBS before being resuspended in 250 μL of DNA staining solution (YEASEN Biotechnology, Shanghai, China). They were allowed to sit in this solution at room temperature for 30 min. Following this, cell cycle analysis was executed utilizing a CytoFLEX Flow Cytometer (BECKMAN COULTER, Brea, CA, USA). The collected data were then processed and analyzed using FlowJo version 10.6.2.

### 4.6. Real-Time Quantitative PCR

Total RNA was extracted using the TRIzol reagent (Invitrogen, Carlsbad, CA, USA), followed by cDNA synthesis facilitated by the Prime Script™ RT reagent Kit (Takara, Maebashi, Japan). The miRNA was isolated and purified using the miRNeasy^®^ Mini kit (QIAGEN, Hilden, Germany) as outlined in the manufacturer’s instructions. Quantitative PCR primers for ACTB, ID1, CyclinD1, FTO, and YTHDF2 were procured from Tsingke Biotechnology (Beijing, China). All primer sequences used are provided in Appendix A. The expression levels of these selected genes were determined in triplicate using the LightCycler^®^96 system (Roche, Rotkreuz, Switzerland). The relative fold changes in expression were computed using the 2^−ΔΔCT^ method.

### 4.7. Western Blotting

Cell lysates were prepared using RIPA buffer (Cell Signaling Technology, Danvers, MA, USA). For Western blotting, the following primary antibodies were utilized: anti-ID1 monoclonal antibody (sc-133104, Santa Cruz, Dallas, TX, USA), anti-CyclinD1 monoclonal antibody (92G2, Cell Signaling Technology), anti-FTO monoclonal antibody (ab126605, Abcam, Cambridge, UK), anti-YTHDF2 polyclonal antibody (FNab09573, FineTest, Guangzhou, China), and anti-β-actin monoclonal antibody (Zsbio, Beijing, China). The immune complexes were subsequently visualized using enhanced chemiluminescence detection reagents (4A BIOTECH, Beijing, China).

### 4.8. IHC Staining

The primary antibodies utilized for IHC staining were identical to those used for the above-mentioned Western blotting. Images from the IHC staining process were captured using an Olympus BX63 microscope (Olympus, Tokyo, Japan).

### 4.9. m^6^A Dot Blot Assay

Total RNA was isolated as previously described. mRNA samples, dissolved in an equal volume of RNA incubation buffer (composition: 20× saline-sodium citrate buffer to deionized formaldehyde in a ratio of 3:2), were denatured at 95 °C for 5 min. The samples were then loaded onto two Amersham Hybond-N+ membranes (GE Healthcare, Chicago, IL, USA). One membrane underwent UV crosslinking for 5 min and was subsequently washed with PBST. This membrane was then stained with 0.02% Methylene blue (Sangon Biotech, Shanghai, China) and scanned to determine the total content of input RNA. For the second membrane, after blocking with 5% non-fat milk, it was incubated overnight at 4 °C with a specific m^6^A antibody (anti-m^6^A monoclonal, 1:1000, 61755 ACTIVE MOTIF). Dot blots were incubated with HRP-conjugated anti-mouse immunoglobulin G (IgG) for 1 h before visualization using BIO-RAD ChemiDoc MP Imaging System(BIO-RAD Laboratories, Hercules, CA, USA).

### 4.10. Methylated RNA Immunoprecipitation (MeRIP)-qPCR

Total RNA was isolated as previously described. The EpiQuik™ CUT&RUN m^6^A RNA Enrichment (MeRIP) Kit (P-9018, EPIGENTEK, Farmingdale, NY, USA) was employed for MeRIP. The process was carried out in accordance with the manufacturer’s instructions. Post-MeRIP, cDNA was produced from both the input and immunoprecipitated RNA fractions, and subsequently analyzed by RT-qPCR as described earlier. The relative m^6^A levels for each transcript were calculated as a percentage of input under each condition, normalized against the respective positive-control m^6^A RNA spike-in, following procedures detailed previously. Fold changes in m^6^A enrichment were computed with control samples normalized to 1.

### 4.11. Animal Studies

NOD/ShiLtJGpt-*Prkdc^em26Cd52^Il2rg^em26Cd22^*/Gpt Coisogenic Genetically Engineered Immunodeficient mice (NCG mice) were purchased from GemPharmatech Co., Ltd. (Nanjing, China). Mice were maintained under specific pathogen-free conditions with a 12-h light/dark cycle and given unrestricted access to tap water and food. Huh7 or Hep3B cells (1 × 10⁶ cells) were resuspended in 200 μL of a 1:1 DMEM: Matrigel (Corning, BD Biosciences, Franklin Lakes, NJ, USA) mixture and subcutaneously injected into NCG mice aged 4 to 6 weeks old. Once the tumors had grown to a size that could be measured, the mice were randomly allocated into two groups (Huh7 n = 5; Hep3B n = 6). They were then intraperitoneally administered with either PBS or 6 mg/kg LDN-212854 twice daily for a period of 10–14 days. The dimensions of the subcutaneous tumors were recorded biweekly. This experimental protocol was sanctioned by the Sichuan University Animal Care and Use Committee and adhered to the guidelines set forth by the Guide for the Care and Use of Laboratory Animals prepared by the National Academy of Sciences.

### 4.12. Statistical Analysis

Analyses including overall survival, Unpaired T test, one-way ANOVA, Wilcoxon Signed Rank test, Fisher’s exact test, and chi-square test were conducted using GraphPad Prism 8.0.1 (GraphPad Software, San Diego, CA, USA). A *p*-value of less than 0.05 was deemed statistically significant.

## 5. Conclusions

Our study has elucidated a crucial mechanism through which the BMP9-ID1 pathway influences the development of HCC. Specifically, it induces CyclinD1 expression and promotes cell cycle progression while reducing m^6^A methylation in an FTO-YTHDF2-dependent manner. These findings suggest that targeting BMP9 signaling in therapy could offer a promising avenue for treating advanced HCC. Further investigation is needed to explore the practical application of BMP receptor inhibitors in clinical settings.

## Figures and Tables

**Figure 1 ijms-25-00981-f001:**
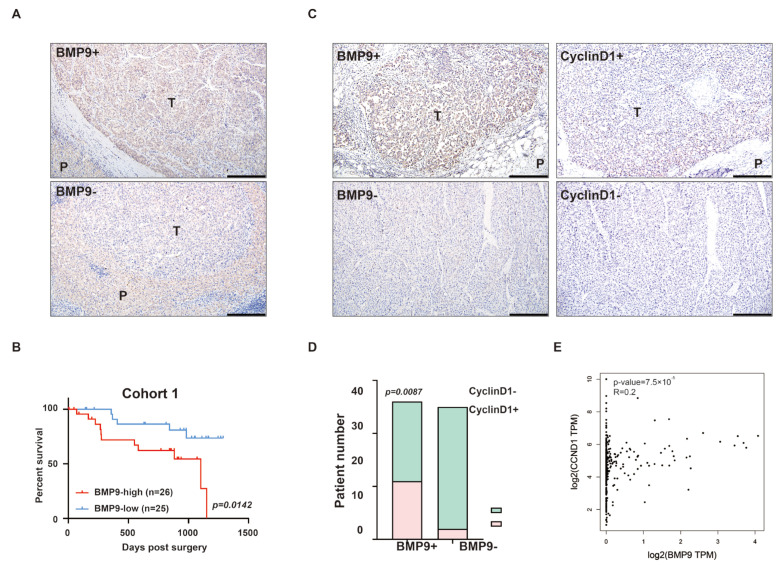
Increased expression of BMP9 is correlated with a poor prognosis in HCC patients and increased CyclinD1 expression in HCC. (**A**) Immunohistochemistry (IHC) analysis of BMP9 in HCC specimens. T = tumor, P = para-tumor. Scale bar = 200 μm (**B**) Kaplan–Meier survival curves according to BMP9 expression in HCC specimens (classified as low or high compare to para-tumor tissue). *p* = 0.0142 (**C**) IHC analysis showing BMP9/CyclinD1 high (BMP9+/CyclinD1+) and BMP9/CyclinD1 low (BMP9-/CyclinD1-) expressions HCC tissue specimens. Scale bar = 200 μm (**D**) Correlation of BMP9 and CyclinD1 in HCC patients. The *p* value was calculated using the chi-squared test. *p* = 0.0087 (**E**) The correlation between BMP9 and CyclinD1 (CCND1) was investigated using data from GEPIA2.

**Figure 2 ijms-25-00981-f002:**
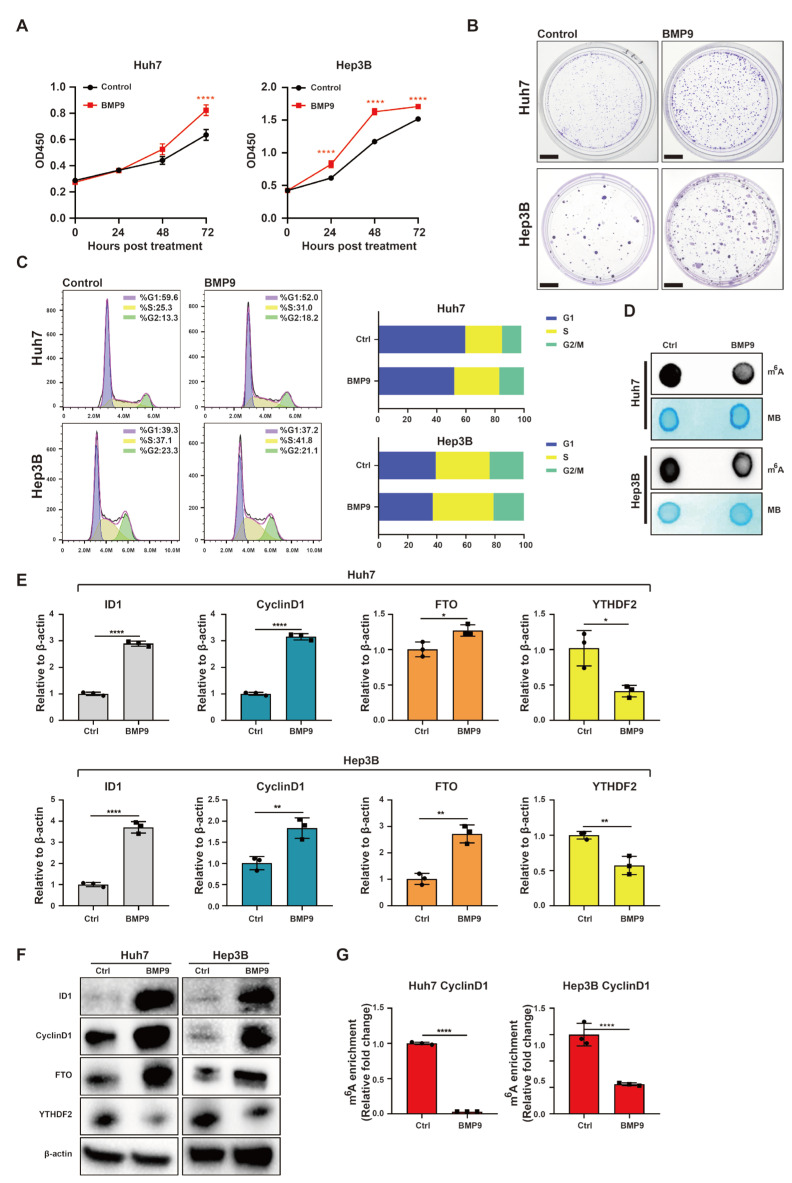
BMP9 enhances CyclinD1 expression in HCC cells to facilitate cell cycle progression via suppressing m^6^A methylation within the 5′ UTR of CyclinD1 mRNA (**A**) CCK-8 assay was used to evaluate the cell proliferation of Huh7 and Hep3B cells. (**B**) Colony formation assay was used to evaluate the cell proliferation of Huh7 and Hep3B cells. Scale bar = 1 cm (**C**) Flow cytometry was used to analyze cell cycle of Huh7 and Hep3B cells. (**D**) m^6^A dot blotting was used to evaluate the global RNA m^6^A methylation of Huh7 and Hep3B cells. (**E**) Relative gene expression levels of ID1, CyclinD1, FTO and YTHDF2 in Huh7 and Hep3B cells. (**F**) Western blot analysis of ID1, CyclinD1, FTO and YTHDF2 in Huh7 and Hep3B cells. (**G**) The m^6^A methylation within the 5′-UTR of CyclinD1 mRNA in Huh7 and Hep3B cells was analyzed using MeRIP-qPCR. Cells underwent treatment with either Dimethyl Sulfoxide (DMSO) or BMP9 (5 ng/mL) for a duration of 48 h. The error bars illustrate the Standard Deviation (SD) derived from a minimum of three independent biological repetitions. Student’s *t*-test was employed to compute the *p*-values, depicted as * *p* < 0.05; ** *p* < 0.01; **** *p* < 0.0001.

**Figure 3 ijms-25-00981-f003:**
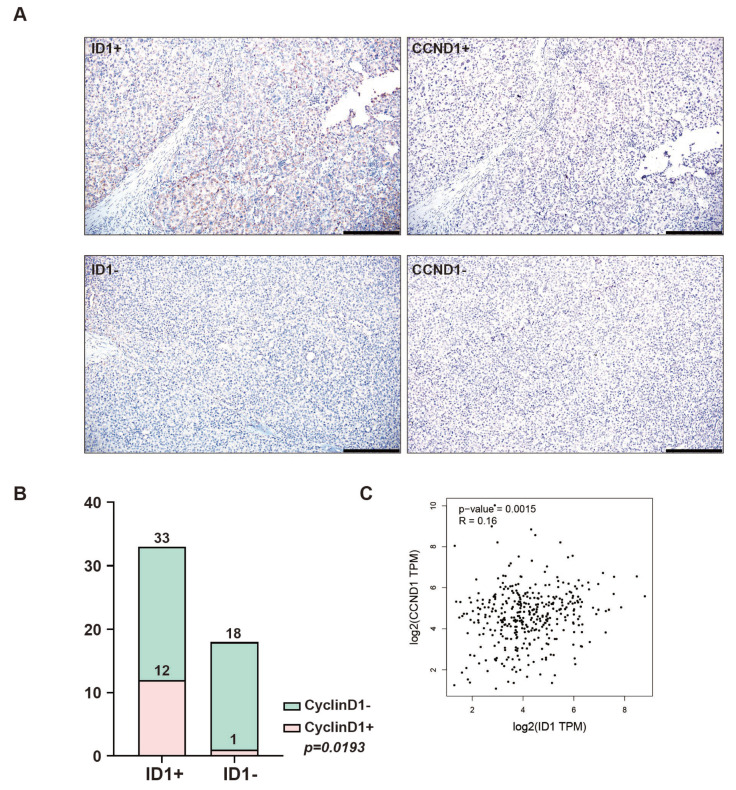
The correlation between ID1 and CyclinD1 in samples of HCC patients. (**A**) IHC analysis of BMP9/CyclinD1 high (BMP9+/CyclinD+) and BMP9/CyclinD1 low (BMP9-/CyclinD-) surgically resected HCC tissue specimens. Scale bar = 200 μm (**B**) Correlation between BMP9 and CyclinD1 in HCC patients. The *p* value was calculated using the chi-squared test. (**C**) The correlation between ID1 and CyclinD1 (*CCND1*) was investigated using data from GEPIA2.

**Figure 4 ijms-25-00981-f004:**
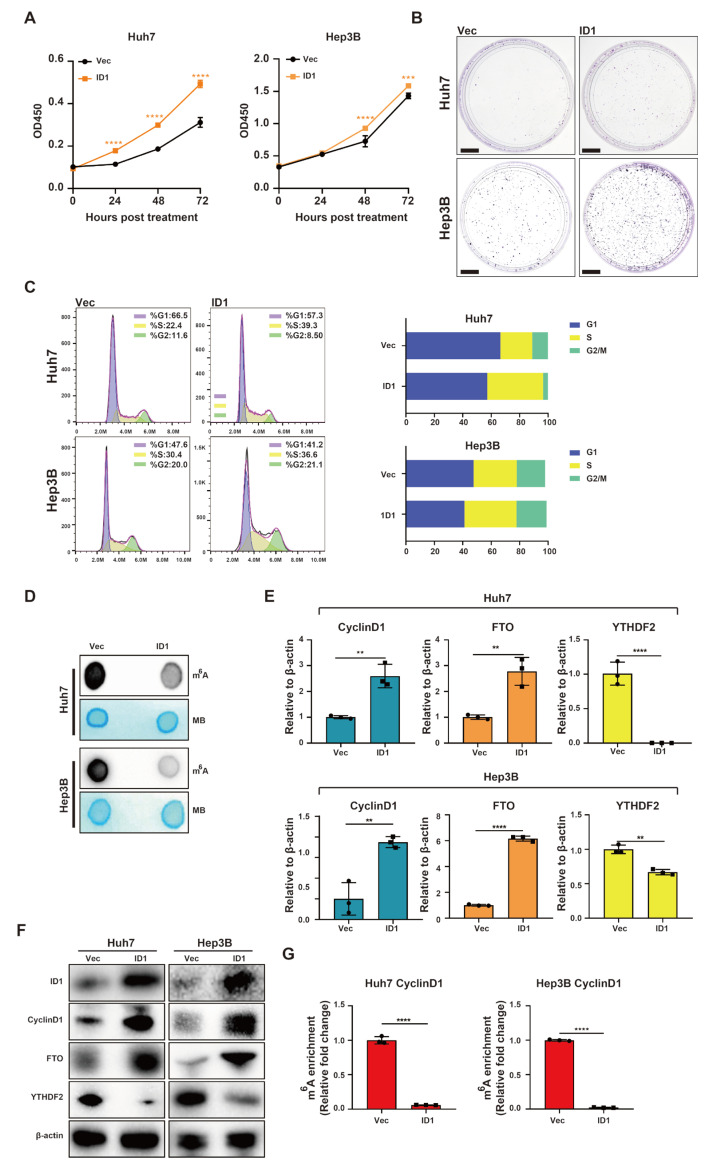
ID1 enhances cell cycle progression and inhibits m^6^A methylation within the 5′ UTR of CyclinD1 mRNA. (**A**) CCK-8 assay was used to evaluate the cell proliferation of Huh7 and Hep3B cells. (**B**) Colony formation assay was used to evaluate the cell proliferation of Huh7 and Hep3B cells. Scale Bar = 1 cm (**C**) Flow cytometry was used to analyze cell cycle of Huh7 and Hep3B cells. (**D**) m^6^A dot blotting was used to evaluate the global RNA m^6^A methylation of Huh7 and Hep3B cells. (**E**) Relative gene expression levels of ID1, CyclinD1, FTO and YTHDF2 in Huh7 and Hep3B cells. (**F**) Western blot analysis of ID1, CyclinD1, FTO and YTHDF2 in Huh7 and Hep3B cells. (**G**) The m^6^A methylation within the 5′-UTR of CyclinD1 mRNA in Huh7 and Hep3B cells was analyzed using MeRIP-qPCR. Cells were transfected 2 μg ID1 plasmid or vector for 48 h. The error bars denote the Standard Deviation (SD) drawn from a minimum of three separate biological replicates. The *p*-values were computed using Student’s *t*-test, represented as ** *p* < 0.01; *** *p* < 0.001; **** *p* < 0.0001.

**Figure 5 ijms-25-00981-f005:**
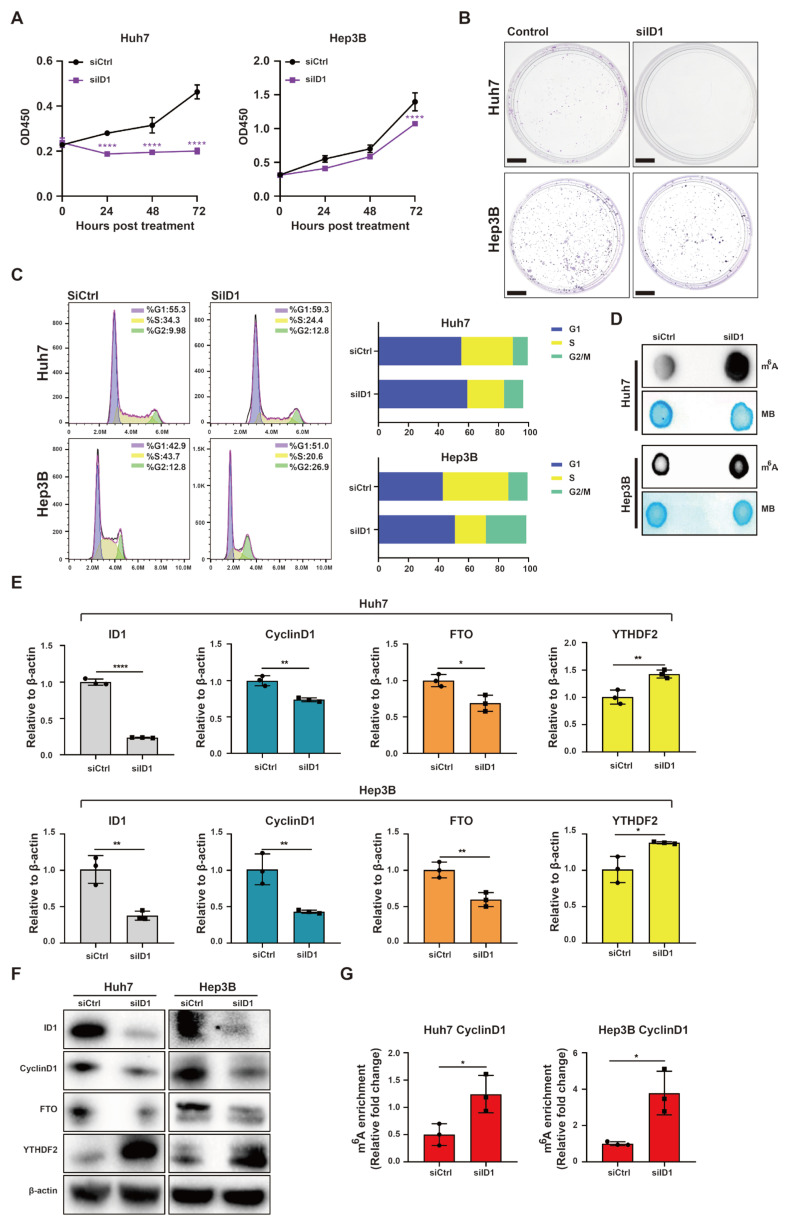
Knockdown of ID1 suppresses cell cycle progression and induces m^6^A methylation within the 5′ UTR of CyclinD1 mRNA. (**A**) CCK-8 assay was used to evaluate the cell proliferation of Huh7 and Hep3B cells. (**B**) Colony formation assay was used to evaluate the cell proliferation of Huh7 and Hep3B cells. Scale bar = 1 cm (**C**) Flow cytometry was used to analyze cell cycle of Huh7 and Hep3B cells. (**D**) m^6^A dot blotting was used to evaluate the global RNA m^6^A methylation of Huh7 and Hep3B cells. (**E**) Relative gene expression levels of ID1, CyclinD1, FTO and YTHDF2 in Huh7 and Hep3B cells. (**F**) Western blot analysis of ID1, CyclinD1, FTO and YTHDF2 in Huh7 and Hep3B cells. (**G**) The m^6^A methylation within the 5′ UTR of CyclinD1 mRNA in Huh7 and Hep3B cells was analyzed using MeRIP-qPCR. Cells were transfected 20 nM ID1 siRNA or siCtrl for 48 h. The error bars denote the Standard Deviation (SD) drawn from a minimum of three separate biological replicates. The *p*-values were computed using Student’s *t*-test, represented as * *p* < 0.05; ** *p* < 0.01; **** *p* < 0.0001.

**Figure 6 ijms-25-00981-f006:**
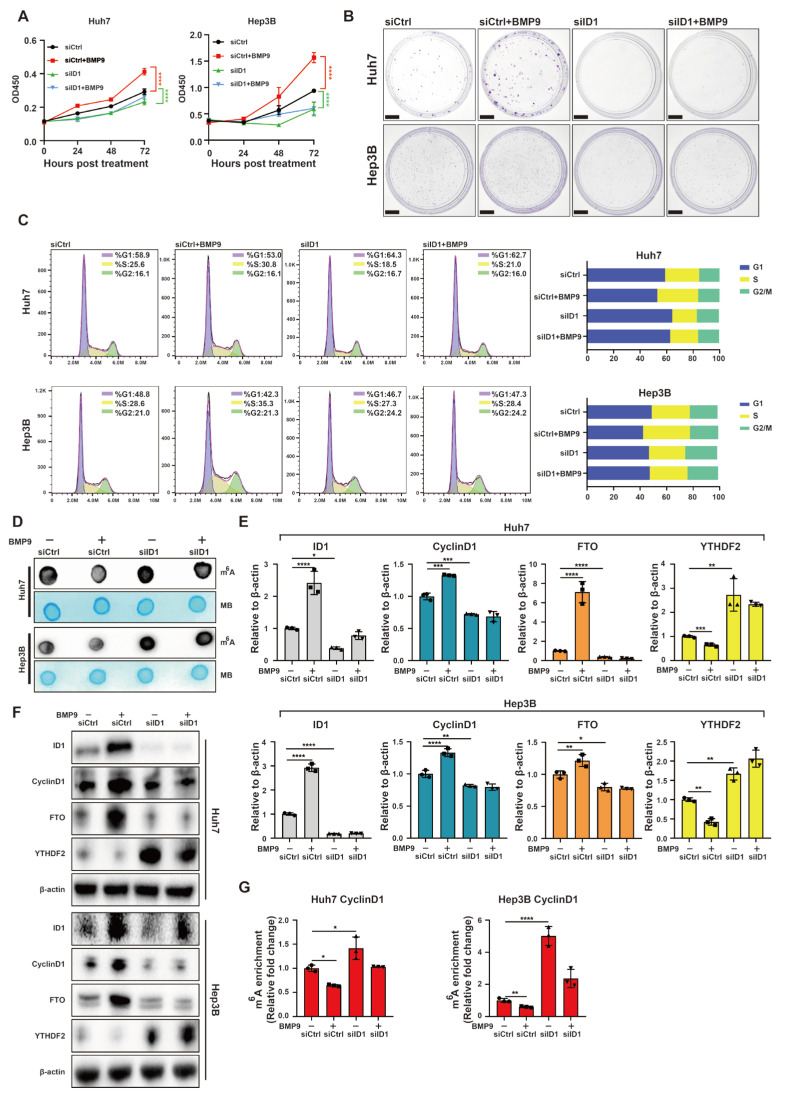
Knockdown of ID1 attenuates the upregulated progression of cell cycle and the downregulated m^6^A methylation within the 5′ UTR of CyclinD1 mRNA induced by BMP9 in HCC cells. (**A**) CCK-8 assay was used to evaluate the cell proliferation of Huh7 and Hep3B cells. (**B**) Colony formation assay was used to evaluate the cell proliferation of Huh7 and Hep3B cells. Scale bar = 1 cm (**C**) Flow cytometry was used to analyze cell cycle of Huh7 and Hep3B cells. (**D**) m^6^A dot blotting was used to evaluate the global RNA m^6^A methylation of Huh7 and Hep3B cells. (**E**) Relative gene expression levels of ID1, CyclinD1, FTO and YTHDF2 in Huh7 and Hep3B cells. (**F**) Western blot analysis of ID1, CyclinD1, FTO and YTHDF2 in Huh7 and Hep3B cells. (**G**) The m^6^A methylation within the 5′ UTR of CyclinD1 mRNA in Huh7 and Hep3B cells was analyzed using MeRIP-qPCR. Cells were treated with BMP9 treated with or without BMP9 (5 ng/mL) for 48 h following siID1 or siCtrl transfection. The error bars denote the Standard Deviation (SD) drawn from a minimum of three separate biological replicates. The *p*-values were computed using Student’s *t*-test, represented as * *p* < 0.05; ** *p* < 0.01; *** *p* < 0.001; **** *p* < 0.0001.

**Figure 7 ijms-25-00981-f007:**
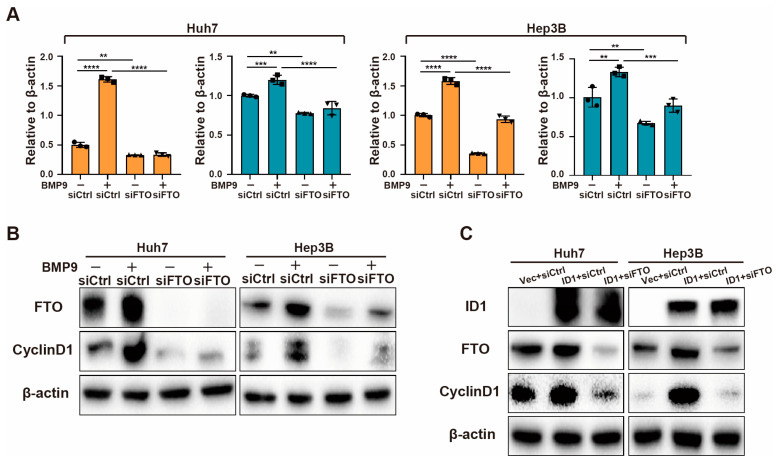
BMP9-ID1 pathway regulates CyclinD1 expression via FTO. (**A**) Relative gene expression levels of FTO and CyclinD1 in Huh7 and Hep3B. Cells were treated with or without BMP9 (5 ng/mL) for 48 h following siRNA transfection. (**B**) Western blot analysis of CyclinD1 and FTO in Huh7 and Hep3B cells. Cells were treated with or without BMP9 (5 ng/mL) for 48 h following siRNA transfection. (**C**) Western blot analysis of CyclinD1 and FTO in Huh7 and Hep3B cells. Cells were co-transfected with ID1 plasmid and siRNA of FTO for 48 h. The error bars denote the Standard Deviation (SD) drawn from a minimum of three separate biological replicates. The *p*-values were computed using Student’s *t*-test, represented as ** *p* < 0.01; *** *p* < 0.001; **** *p* < 0.0001.

**Figure 8 ijms-25-00981-f008:**
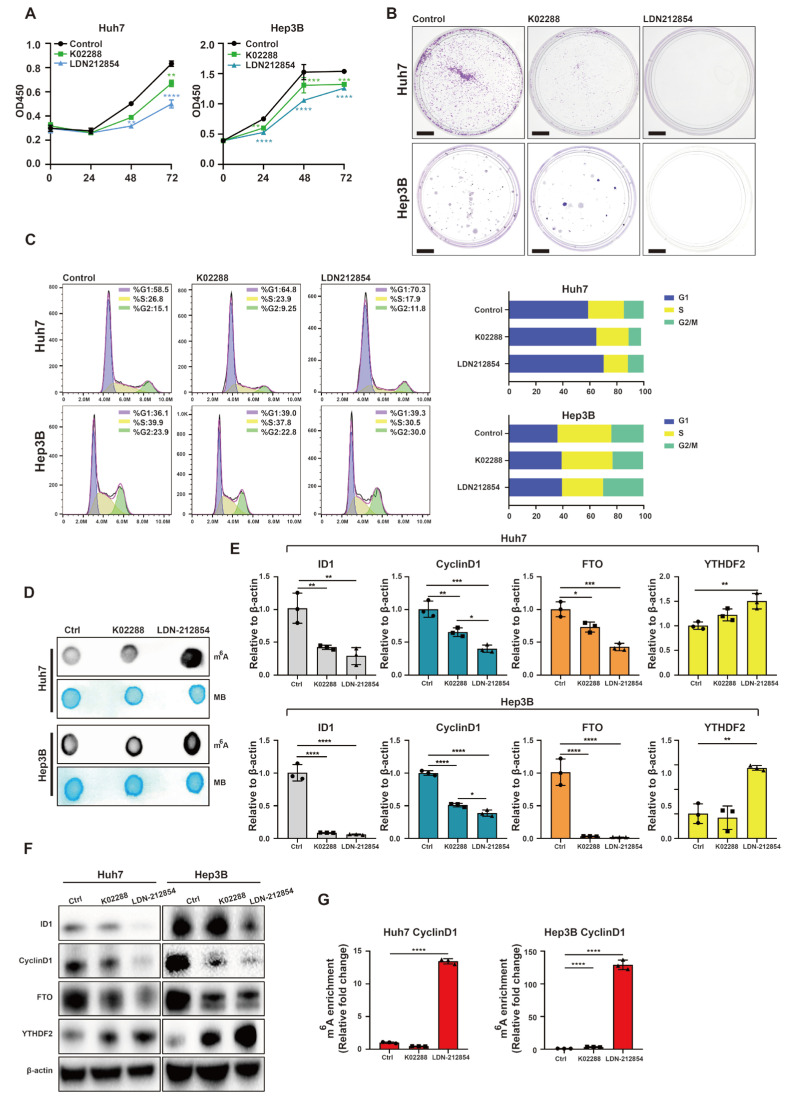
BMP receptor inhibitors repress cell cycle progression and promote m^6^A methylation within the 5′ UTR of CyclinD1 mRNA in HCC cells. (**A**) CCK-8 assay was used to evaluate the cell proliferation of Huh7 and Hep3B cells. (**B**) Colony formation assay was used to evaluate the cell proliferation of Huh7 and Hep3B cells. Scale bar = 1 cm (**C**) Flow cytometry was used to analyze cell cycle of Huh7 and Hep3B cells. (**D**) m^6^A dot blotting was used to evaluate the global RNA m^6^A methylation of Huh7 and Hep3B cells. (**E**) Relative gene expression levels of ID1, CyclinD1, FTO and YTHDF2 in Huh7 and Hep3B cells. (**F**) Western blot analysis of ID1, CyclinD1, FTO and YTHDF2 in Huh7 and Hep3B cells. (**G**) The m^6^A methylation within the 5′-UTR of CyclinD1 mRNA in Huh7 and Hep3B cells was analyzed using MeRIP-qPCR. Cells were treated with 2 μM K02288, LDN-212854 or DMSO for 48 h. The error bars denote the Standard Deviation (SD) drawn from a minimum of three separate biological replicates. The *p*-values were computed using Student’s *t*-test, represented as * *p* < 0.05; ** *p* < 0.01; *** *p* < 0.001; **** *p* < 0.0001.

**Figure 9 ijms-25-00981-f009:**
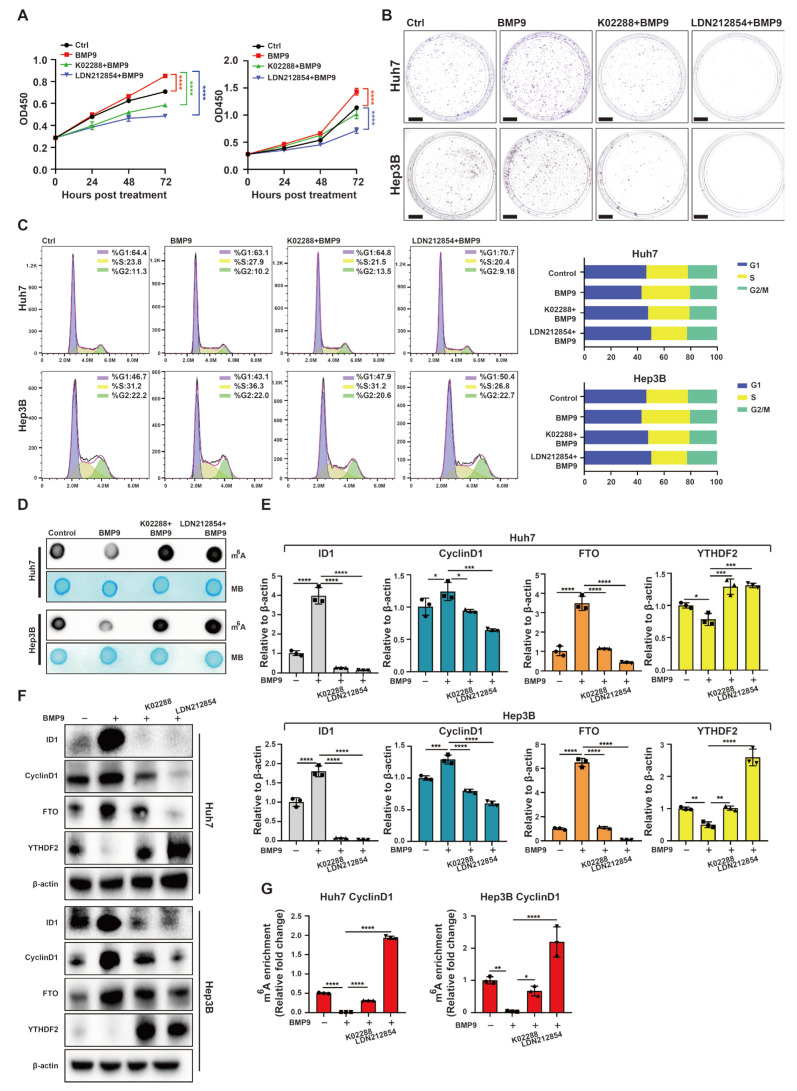
BMP receptor inhibitors attenuate upregulated cell cycle progression and downregulated m^6^A methylation within 5′ UTR of CyclinD1 mRNA induced by BMP9 in HCC cells. (**A**) CCK-8 assay was used to evaluate the cell proliferation of Huh7 and Hep3B cells. Scale bar = 1 cm (**B**) Colony formation assay was used to evaluate the cell proliferation of Huh7 and Hep3B cells. (**C**) Flow cytometry was used to analyze cell cycle of Huh7 and Hep3B cells. (**D**) m^6^A dot blotting was used to evaluate the global RNA m^6^A methylation of Huh7 and Hep3B cells. (**E**) Relative gene expression levels of ID1, CyclinD1, FTO and YTHDF2 in Huh7 and Hep3B cells. (**F**) Western blot analysis of ID1, CyclinD1, FTO and YTHDF2 in Huh7 and Hep3B cells. (**G**) The m^6^A methylation within the 5′ UTR of CyclinD1 mRNA in Huh7 and Hep3B cells was analyzed using MeRIP-qPCR. Cells were treated with 2 μM K02288, LDN-212854 or DMSO in the presence of BMP9 (5 ng/mL) for 48 h. The error bars denote the Standard Deviation (SD) drawn from a minimum of three separate biological replicates. The *p*-values were computed using Student’s *t*-test, represented as * *p* < 0.05; ** *p* < 0.01; *** *p* < 0.001; **** *p* < 0.0001.

**Figure 10 ijms-25-00981-f010:**
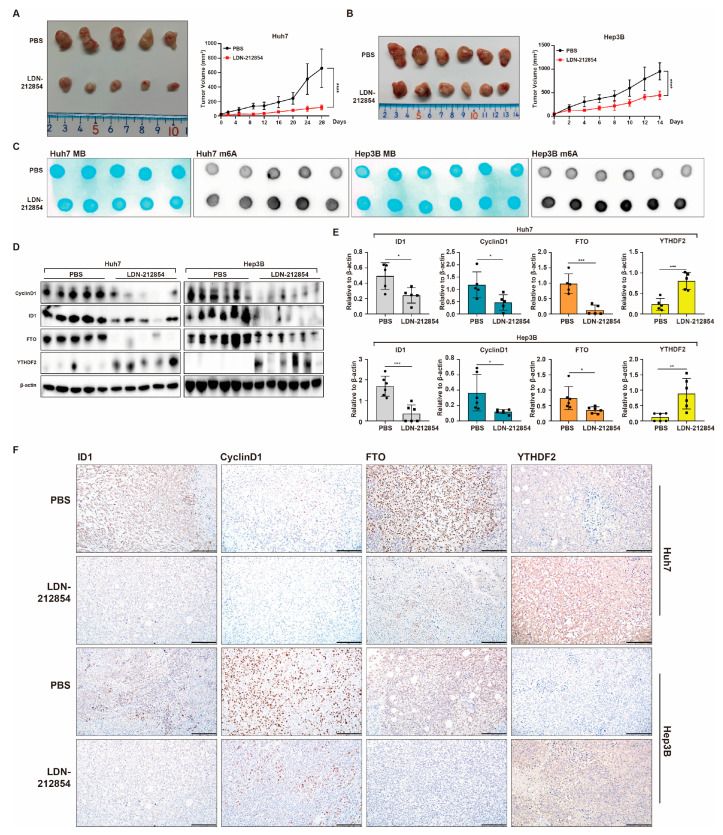
BMP receptor inhibitor LDN-212854 represses tumor growth and promotes global RNA m^6^A methylation of HCC xenografts. (**A**) Influence of LDN-212854 on growth of Huh7 xenograft tumors. Mice bearing Huh7 xenograft tumors were treated with PBS (n = 5) or LDN-212854 (n = 5). (**B**) Influence of LDN-212854 on growth of Hep3B xenograft tumors. Mice bearing Hep3B xenograft tumors were treated with PBS (n = 6) or LDN-212854 (n = 6). (**C**) m^6^A dot blotting showed the global RNA m^6^A methylation in Huh7 and Hep3B xenograft tumors. (**D**,**E**) Western blot analysis of CyclinD1, ID1, FTO and YTHDF2 expressions in Huh7 and Hep3B xenograft tumors. β-actin was used as the reference for quantifying protein expression. (**F**) IHC analysis of ID1, CyclinD1, FTO and YTHDF2 expression in Huh7 and Hep3B xenograft tumors. Scale bar = 200 μm. The error bars denote the Standard Deviation (SD) drawn from a minimum of three separate biological replicates. The *p*-values were computed using Student’s *t*-test, represented as * *p* < 0.05; ** *p* < 0.01; *** *p* < 0.001; **** *p* < 0.0001.

**Figure 11 ijms-25-00981-f011:**
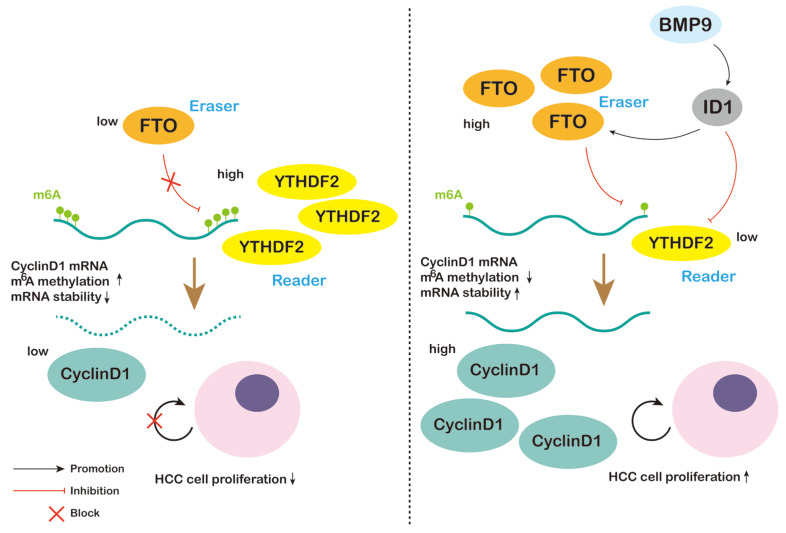
Schematic of BMP9-ID1 pathway promoting HCC cell proliferation by attenuating the m^6^A methylation within 5′ UTR of CyclinD1 mRNA.

**Table 1 ijms-25-00981-t001:** Clinical characteristics of HCC patients in BMP9- high and low groups in Figure 1B.

Parameter	BMP9-High (n = 26)	BMP9-Low (n = 25)	*p* Value *
Age (years, mean, SEM)	53.38, 1.981	54.32, 1.892	0.7345
Sex (M/F)	23/3	22/3	0.7013
AFP (ng/mL, median)	262.5	8.06	0
PIVKA-II (ng/mL, mean, SEM)	9230, 3763	4743, 2173	0.3118
BCLC stage (0–B, C–D)	16, 10	21, 4	0.1381
Liver cirrhosis (F1–F2/F3–F4)	2/24	3/22	0.9632
Microscopic PV invasion (yes/no)	12/14	12/13	0.8949
Tumor size (cm^3^, mean, SEM)	239, 79.62	106.5, 34.08	0.1377
Recurrence (yes/no)	15/11	10/15	0.2064
Recurrence pattern (single/multiple/extrahepatic)	2/9/4	1/5/4	0.7803
Metastasis (yes/no)	13/13	8/17	0.1917
Metastasis pattern (intrahepatic/extrahepatic)	10/3	4/4	0.3458

AFP, alpha-fetoprotein; PIVKA-II, protein induced by vitamin K absence or antagonist-II; PV, portal vein; BCLC, Barcelona clinic liver cancer. * Unpaired T test (Age, PIVKA-II, Tumor size), Wilcoxon Signed Rank test (AFP), Fisher’s exact test (Liver cirrhosis), or chi-square test (sex, BCLC stage, Liver cirrhosis, microscopic PV invasion, Recurrence, Recurrence Pattern, Metastasis, Metastasis Pattern).

## Data Availability

The GEPIA2 dataset is publicly available. The data that support the findings of this study are available in the relevant figures/tables of this article.

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
