# Peer review of "BMP9-ID1 Pathway Attenuates N6-Methyladenosine Levels of CyclinD1 to Promote Cell Proliferation in Hepatocellular Carcinoma"

_ijms, 2024, doi:10.3390/ijms25020981_

Round 1
Reviewer 1 Report
Comments and Suggestions for Authors
This is a well organized paper. The authors target on the BMP9-ID1 pathway for the N6-Methyladenosine levels analysis. Here are some major concerns.
1. Figure 1 and 3:
a. How did the author binarize the BMP9 expression and CCND1 expression as high and low or positive vs negative? On Figure 1E, their expression levels are continuous. How to choose the cutoff and why? Same concern for gene ID1 in Figure 3.
b. Figure 1B: no x and y labels
c. Figure 1E: what statistical test did the author apply to generate the p-value? Given the R value is low, why the p-value looks very small? Similar question for Figure 3C, what’s the statistical test that has been applied?
d. Table 2 and Figure 1D are duplicate. Table 3 and Figure 3B are duplicate. The authors can put the values from the TAble on the top of the figure.
2. Figure 2, 4 and 5:
a. Figure 2A, 4A, 5A, 6A, 8A, 9A: no x label
b. Figure 6A: the four color lines are not well annotated on the figure.
c. Figure 2E, 4E, 5E, 6E, 7A, 8E, 9E: how many repeats per condition? Please plot each individual points on the top of your bar graph.
3. The m6A analysis is very targeted. There are techniques such as m6A-seq for unbiased methylation calling on the whole transcriptomes. What’s the rationale to performed targeted analysis, rather than an unbiased search?
Comments on the Quality of English Language
Great English presentation.
Author Response
Dear Reviewer 1: All the places that have been modified are marked with red text. Please see the attachment

Reviewer 2 Report
Comments and Suggestions for Authors
The work “BMP9-ID1 signaling attenuates N6 -Methyladenosine levels of 2 CyclinD1 to promote cell proliferation in hepatocellular carcinoma” is quite interesting, but a number of issues have arisen that make it impossible to rate the study highly. In order to be published, a paper must pass a major revision.
1) Abstract:L. 16 – “Our goal was to investigate the role of BMP9 signaling in regulating N6-methyl- adenosine (m6A) methylation and cell cycle progression, and evaluate the therapeutic potential of BMP receptor inhibitors for HCC treatment.Выражение “BMP9 reduced m6A methylation” некорректно, далее уточняется : “BMP9-ID1 signaling promotes HCC cell proliferation by 25 down-regulating m6A methylation of CyclinD1” - The authors are referring to methylation of adenosine or N6-methyl- adenosine (addition methylation?)?
2) Please, indicate in the abstract what methods were used to analyze the level of RNA methylation?
3) Please, indicate in the abstract what cancer models/cell model were used for analysis.
4. Please, add to the Introduction what is known about the biological function of RNA methylation. Which RNA classes are most characterized by this modification (mRNA, tRNA etс. )?
5. In the Results part, it should be clearly stated what methylation level was investigated. Especially this information is missing in section 2.2. L. 125 - 128 “Moreover, we measured the global m6A levels in both the negative control and BMP9-treated groups in HCC cell lines using m6A dot blotting. Remarkably, treatment with BMP9 led to a substantial reduction in m6A levels in both HCC cell lines (Figure 2D).”
6. Since the authors show that BMP9 influence the cell cycle, it is necessary to provide data on the dynamics of mRNA levels (proteins associated with m6A methylation modification) for at least 48 h of culturing via RT-qPCR.
Author Response
Dear Reviewer 2: All the places that have been modified are marked with red text. Please see the attachment.

Round 2
Reviewer 1 Report
Comments and Suggestions for Authors
Thanks for the revision work done by the authors, which have addressed most of my concerns.
Author Response
Dear Reviewer,
Thank you very much. I appreciate the valuable feedback you provided on my manuscript. Your insights have greatly enhanced the quality of my work. I eagerly anticipate your further review and input.
Reviewer 2 Report
Comments and Suggestions for Authors
The authors have significantly improved the manuscript. In its current form it can be published.
Author Response
Dear Reviewer,
Thank you for your valuable comments on my manuscript. Your suggestions have significantly improved the quality of my manuscript. I look forward to your review once again.